# Koumine Promotes ROS Production to Suppress Hepatocellular Carcinoma Cell Proliferation Via NF-κB and ERK/p38 MAPK Signaling

**DOI:** 10.3390/biom9100559

**Published:** 2019-10-02

**Authors:** Zhihang Yuan, Zengenni Liang, Jine Yi, Xiaojun Chen, Rongfang Li, Jing Wu, Zhiliang Sun

**Affiliations:** 1Department of Clinical Veterinary Medicine, College of Veterinary Medicine, Hunan Agricultural University, Changsha 410128, China; zhyuan2016@hunau.edu.cn (Z.Y.); yijine@gmail.com (J.Y.); s51857176@gmail.com (X.C.); 2019nongda@gmail.com (R.L.); 2Hunan Co-Innovation Center for Utilization of Botanical Functional Ingredients, Changsha 410128, China; 3Department of Hunan Agricultural Product Processing Institute, Changsha 410128, China; 2019enni@gmail.com

**Keywords:** hepatocellular carcinoma (HCC), koumine, ROS, proliferation, NF-κB and ERK/p38 MAPK signaling

## Abstract

In the past decades, hepatocellular carcinoma (HCC) has been receiving increased attention due to rising morbidity and mortality in both developing and developed countries. Koumine, one of the significant alkaloidal constituents of *Gelsemium elegans* Benth., has been regarded as a promising anti-inflammation, anxiolytic, and analgesic agent, as well as an anti-tumor agent. In the present study, we attempted to provide a novel mechanism by which koumine suppresses HCC cell proliferation. We demonstrated that koumine might suppress the proliferation of HCC cells and promote apoptosis in HCC cells dose-dependently. Under koumine treatment, the mitochondria membrane potential was significantly decreased while reactive oxygen species (ROS) production was increased in HCC cells; in the meantime, the phosphorylation of ERK, p38, p65, and IκBα could all be inhibited by koumine treatment dose-dependently. More importantly, the effects of koumine upon mitochondria membrane potential, ROS production, and the phosphorylation of ERK, p38, p65, and IκBα could be significantly reversed by ROS inhibitor, indicating that koumine affects HCC cell fate and ERK/p38 MAPK and NF-κB signaling activity through producing excess ROS. In conclusion, koumine could inhibit the proliferation of HCC cells and promote apoptosis in HCC cells; NF-κB and ERK/p38 MAPK pathways could contribute to koumine functions in a ROS-dependent manner.

## 1. Introduction

Hepatocellular carcinoma (HCC) is the most commonly seen primary liver cancer; it accounts for 85% of the liver cancers [1]. Other forms include cholangiocarcinoma, which begins with cells surrounding the bile duct, angiosarcoma (or hemangiosarcoma), and hepatoblastoma. Mainly occurring in the developing world [2,3], HCC has been receiving increased attention due to rising morbidity and mortality in many countries for the past few years [4,5,6].

Environmental factors or cellular mitochondrial dysfunction lead to the production of reactive oxygen species (ROS), thus resulting in sustained oxidative stress, which is related to liver carcinogenesis according to recent studies [7]. Reduced ROS is required for cell proliferation, apoptosis, cell cycle arrest, cell senescence, and other physiological processes [8]. Nevertheless, improved ROS can induce oxidative stress and provide an environment that is potentially toxic to cells. Many intracellular and extracellular factors can cause endoplasmic reticulum (ER)-stress, a disorder of Ca^2+^ homeostasis and mitochondrial ROS production resulting in apoptosis [9,10,11,12]. In addition, it has been revealed by proteomics results that increased NF-κB-related Wnt-1 expression is a critical mechanism for liver carcinogenesis [13]. MAPK cascade can transduce signals from tyrosine kinase receptors, such as vascular endothelial growth factor receptor (VEGFR), epidermal growth factor receptor (EGFR), insulin-like growth factor receptor (IGFR), and hepatocyte growth factor receptor (HGFR). Within the cascade, activated Ras (Ras-GTP) could induce the activation of RAF-1, MEK-1/2, and ERK-1/2 in sequence. ERK1/2 can enter into the nucleus after activation or phosphorylation, where it transactivates c-JUN, c-FOS, c-MYC (contributing to the mechanisms of cell proliferation and survival), VEGF and HIF1α (modulating angiogenesis), hexokinase II, and other growth-associated genes [14,15,16]. Even without growth factors, ERK1/2 constitutive activation leads to increased cell proliferation, thus resulting in tumor development.

Koumine, one of the significant alkaloidal constituents of *Gelsemium elegans* Benth., has increasingly received greater attention because of its multiple biological effects [17]. Koumine has been regarded as a promising anti-inflammation, anxiolytic, and analgesic agent, as well as an anti-tumor agent [18,19,20,21]. Koumine exerts its biological functions in tumors by modulating different intracellular physiological processes via diverse mechanisms. In human breast cancer cells, koumine promotes apoptosis and cell cycle arrest in G2/M phase via reducing Bcl2 and increasing the pro-apoptotic factors Bax and Caspase-3 [22]. In human colonic adenocarcinoma cells, koumine can inhibit the mitochondrial membrane potential while enhancing the production of ROS [23]. Within human cervical cancer HeLa cells, studies have found that koumine promotes the apoptosis and cycle arrest of cancer cells by suppressing ROS-dependent NF-κB pathway [24]. Interestingly, koumine reduces proinflammatory factor production within mouse macrophages via inhibiting ERK/p38 MAPK phosphorylation and the NF-κB pathway [25]. Considering the critical roles of ROS and ERK/p38 MAPK and NF-κB signaling pathways within HCC, we hypothesize that koumine contributes to regulating the signaling pathways of NF-κB and ERK/p38 MAPK within HCC through the excessive production of ROS, therefore inhibiting HCC cell proliferation and promoting HCC cell apoptosis.

Herein, the killing effects of koumine upon HCC were evaluated by examining HCC cell viability, apoptosis, and apoptosis-related factors. Next, the changes in the mitochondrial membrane potential, ROS production, and ERK/p38 MAPK and NF-κB pathways in response to koumine treatment were determined. Finally, the dynamic effects of koumine and ROS inhibitor on HCC cells were examined to investigate whether koumine exerts its effects via ROS production and ERK/p38 MAPK and NF-κB signaling pathways. These data indicate that koumine exerts effects upon HCC cell proliferation and apoptosis and shed light on the underlying mechanism. According to the findings of this research, koumine might be a promising anti-tumor agent for HCC treatment.

## 2. Materials and Methods

### 2.1. Cell Lines and Cell Culture

Huh-7 cell line (JCRB0403) was obtained from the Japanese Collection of Research Bioresources Cell Bank (Osaka, Japan) and cultured in Dulbecco’s minimal essential medium (DMEM) with 10% fetal bovine serum (FBS) (Invitrogen, Waltham, MA, USA). SNU-449 cell line (ATCC CRL-2234) was obtained from ATCC (Manassas, VA, USA) and cultured in RPMI-1640 Medium (Catalog No. 30-2001; ATCC) supplemented with 10% FBS. All cells were cultured at 37 °C in 5% CO_2_. For koumine and N-acetylcysteine (NAC) treatment, HCC cells were exposed to different concentration of koumine (100 μg/mL, 200 μg/mL, 400 μg/mL, and 800 μg/mL) or 400 μg/mL koumine plus 800 μM NAC for 24 h, then cells were harvested for further experiments.

### 2.2. Cell Viability Determined by MTT Assays

The cell viability was determined by a modified MTT assay following previously described methods [26]. After discarding the supernatant, the formazan was dissolved by DMSO; then, the optical density (OD) values were determined at 490 nm. The cell viability was calculated by taking the cell viability in the non-treatment group as 100%.

### 2.3. Cell Apoptosis Determined by Flow Cytometry

The cell apoptosis was determined using flow cytometry by using Cell Apoptosis Kit with Annexin V-FITC & Propidium Iodide (PI) (Nanjing KeyGen Biotech, Nanjing, China) following previously described [25]. Data procession was conducted by Flow Cytometry analysis (BD, New York, NY, USA).

### 2.4. Immunoblotting

Protein concentrations of cleaved-Caspase3, Caspase3, Bax, Bcl-2, cytochrome c, p-ERK, ERK, p-p38, p38, p-p65, p65, p-IκBα, and IκBα were quantified using the BCA kit (Beyotime, Shanghai, China) and then the protein levels were determined following previously methods described [27] using the antibodies listed below: anti-cleaved-Caspase3 (ab2302, Abcam, Cambridge, MA, USA), anti-Caspase3 (ab13847, Abcam), anti-Bax (ab32503, Abcam), anti-Bcl-2 (ab32124, Abcam), anti-cytochrome c (ab13575, Abcam), anti-p-ERK (ab50011, Abcam), anti-ERK (ab54230, Abcam), anti-p-p38 (ab31828, Abcam), anti-p38 (ab4822, Abcam), anti-p65 (ab16502, Abcam), anti-p-p65 (ab86299, Abcam), anti-p-IκBα (#2859, Cell Signaling, Danvers, MA, USA), anti-IκBα (#2859, Cell Signaling), anti-Tubulin (ab6046, Abcam), anti-GAPDH (ab8245, Abcam), anti-COXIV (ab14744, Abcam), and horseradish peroxidase combined second antibody. The binding antibody was visualized with an enhanced chemiluminescence detection system (ECL) (Beyotime, Shanghai, China). GAPDH was used as an endogenous control. Tubulin was used for cytoplasm protein loading control. COXIV was used as mitochondrial protein loading control.

### 2.5. Mitochondrial Membrane Potential (ΔΨm) Assay

ΔΨm was detected in Huh-7 and SNU-449 cells with the JC-1 mitochondrial transmembrane potential detection kit (Beyotime, Shanghai, China) according to the manufacturer’s protocols and the methods described previously [28].

### 2.6. Determination of the Intracellular ROS

The intracellular ROS levels were determined by using a ROS Assay Kit (Beyotime, Shanghai, China) following the protocols and the methods described previously [29]. Target cells were treated under different conditions and incubated with 2’,7’-Dichlorodihydrofluorescein diacetate (DCFH-DA) for thirty minutes at 37 °C. Then cells were harvested for flow cytometery (BD, USA) analysis [30,31].

### 2.7. Data Process and Statistical Analysis

All the data obtained from at least three independent experiments were processed using GraphPad Prism 5 software (San Diego, CA, USA) and presented as the mean ± standard deviation (SD). One-way analysis of variance (ANOVA) was used for all data analyses. A *p* value of < 0.05 was considered statistically significant.

## 3. Results

### 3.1. The Killing Effects of Koumine upon Hepatocellular Carcinoma Cells

Firstly, we treated HCC cells with 0 μg/mL, 100 μg/mL, 200 μg/mL, 400 μg/mL, and 800 μg/mL koumine and examined the specific cellular functions. As revealed by the MTT assays, the cell viability of HCC cells were significantly inhibited by koumine dose-dependently (Figure 1A). Meanwhile, it was demonstrated by flow cytometry that the apoptosis was enhanced by koumine dose-dependently (Figure 1B). Consistently, apoptosis-associated factor protein levels, including cleaved-Caspase3 and Bax, were significantly increased, while Caspase3 and Bcl2 protein levels were decreased by koumine dose-dependently (Figure 1C). In summary, koumine might affect the proliferation and apoptosis of HCC cells dose-dependently.

### 3.2. Koumine Induced Mitochondrial Dysfunction and ROS Production in HCC Cells

It was found that, in human colorectal adenocarcinoma cells, koumine can reduce mitochondrial membrane potential within cancer cells while increasing ROS production [3]. Thus, the effects of koumine upon the mitochondrial function and ROS production in HCC cells were determined. We treated HCC cells with 0 μg/mL, 400 μg/mL, and 800 μg/mL koumine and examined related indicators. Koumine treatment significantly decreased the mitochondrial membrane potential while it increased the ROS production dose-dependently in HCC cells (Figure 2A,B). Moreover, cytochrome C protein level was determined within the cytoplasm and mitochondria of HCC upon 0, 400, and 800 μg/mL koumine treatment; as shown in Figure 2C, cytochrome C protein could be remarkably upregulated within cytoplasm while it was downregulated within mitochondria by koumine treatment dose-dependently. In summary, koumine treatment could modulate the mitochondrial functions and ROS production of HCC cells.

### 3.3. Koumine Inhibits NF-κB and ERK/p38 MAPK Signaling Pathways within HCC Cells

To further confirm the molecular mechanism, we also examined the effects of koumine upon NF-κB and ERK/p38 MAPK signaling pathways. We treated HCC with 0 μg/mL, 400 μg/mL, and 800 μg/mL koumine and examined p-ERK, ERK, p-p38, p38, p-p65, p65, p-IκBα, and IκBα protein levels. Figure 3 shows that ERK, p38, p65, and IκBα phosphorylation could be remarkably decreased by 400 and 800 μg/mL koumine, and further reduced by 800 μg/mL koumine treatment.

### 3.4. Koumine Modulated HCC Cell Apoptosis and ERK/p38 MAPK and NF-κB Signaling Activation via Producing Excessive ROS

To further investigate the role of koumine-induced excessive ROS generation, HCC cells were co-treated with 400 μg/mL koumine and 800 μM ROS inhibitor (NAC) and examined for the related indicators. Koumine induced the cell viability inhibition, and ROS generation was significantly reversed by NAC co-treatment (Figure 4A,B). Moreover, the cell apoptosis rate was also significantly inhibited by NAC co-treatment (Figure 4C).

Next, the western blot results showed that p-ERK, p-p38, p-p6, and p-IκBα protein levels were decreased by koumine, while partially reversed by NAC co-treatment, without affecting the total ERK, p38, p65 and IκBα protein (Figure 5A). Koumine-increased cleaved-Caspase 3 and Bax protein levels were also partially reduced by NAC co-treatment. In the contrast, caspase-3 and Bcl2 protein levels were increased by NAC co-treatment (Figure 5B). These data indicate that koumine-induced cell apoptosis and inhibition of ERK/p38 MAPK and NF-κB signaling were associated with excessive ROS generation.

## 4. Discussion

We demonstrated that koumine might suppress the proliferation of HCC cells while promoting apoptosis in HCC cells dose-dependently. Under koumine treatment, the mitochondria membrane potential was significantly decreased, while ROS production was increased in HCC cells; in the meantime, the phosphorylation of ERK, p38, p65, and IκBα could all be inhibited by koumine treatment dose-dependently. More importantly, the effects of koumine upon mitochondria membrane potential, ROS production, and the phosphorylation of ERK, p38, p65, and IκBα could be significantly reversed by ROS inhibitor, indicating that koumine affects HCC cell fate through producing excess ROS via ERK/p38 MAPK phosphorylation and NF-κB signaling.

The anti-tumor effects of koumine on many types of cancers have been widely reported previously [32,33]. As for the underlying mechanism, the anti-tumor effects of koumine have been attributed to its effects on mitochondria functions and mitochondrial production of ROS. Koumine not only lowers colorectal cancer LoVo cell membrane potential and mitochondrial membrane potential but also frees cytosolic calcium concentration, while it increases ROS production and LoVo cell gap junction intercellular communication. Via the above-mentioned mechanisms, koumine induces LoVo cell apoptosis [23,33]. In breast cancer, koumine affected the apoptotic Caspase 3/Bcl-2 cascades to induce G2/M arrest and apoptosis in breast cancer MCF-7 cells [22]. Herein, koumine can remarkably suppress the proliferation of HCC cells while it promotes the apoptosis of HCC by enhancing cleaved-Caspase 3 and Bax protein levels, whereas it inhibits Bcl-2 protein. Additionally, koumine treatment also dose-dependently reduced the mitochondria membrane potential and increased the production of ROS, indicating that the killing effects of koumine upon HCC cells could be attributed to koumine-induced excessive production of ROS by mitochondria.

An essential feature for cancer cells is the possibility to respond to various proliferative or inflammatory factors provided by the microenvironment, possibly through several essential signaling pathways. For example, MAPK signaling pathways, which are involved in mediating processes of cell growth, survival, and death, could be activated in response to various chemicals and environmental stresses [34,35] and then induce apoptosis by phosphorylating or indirectly down-regulating pro-survival Bcl-2 proteins under conditions of cellular stress [36,37]. Earlier publications about HCC have often focused on activation of the ERK pathway by serum factors [38] and inhibition of ERK phosphorylation by sorafenib, a multikinase inhibitor and one of the most widely used anti-tumor agents for HCC treatment. Additionally, another anti-tumor agent, evodiamine (Evo), an active ingredient isolated from the fruit of *Evodia rutaecarpa* Bentham, has been shown to exert its antitumor activities via inhibiting the activation of NF-κB and MAPK [39,40,41]. In the present study, we also investigated the involvement of NF-κB and MAPK signaling pathways in the anti-tumor effects of koumine on HCC. Koumine treatment dramatically inhibited ERK, p38, p65, and IκBα phosphorylation in HCC cells. More importantly, the inhibitory effects of koumine on the phosphorylation of these factors mentioned above could be significantly reversed by the application of ROS inhibitor, indicating that koumine exerts its effects on HCC cells through NF-κB and ERK/p38 MAPK pathways ROS-dependently (Figure 6).

## 5. Conclusions

In conclusion, koumine was shown to inhibit the proliferation and promote the apoptosis in HCC cells; NF-κB and ERK/p38 MAPK pathways were shown to contribute to koumine functions in a ROS-dependent manner.

## Figures and Tables

**Figure 1 biomolecules-09-00559-f001:**
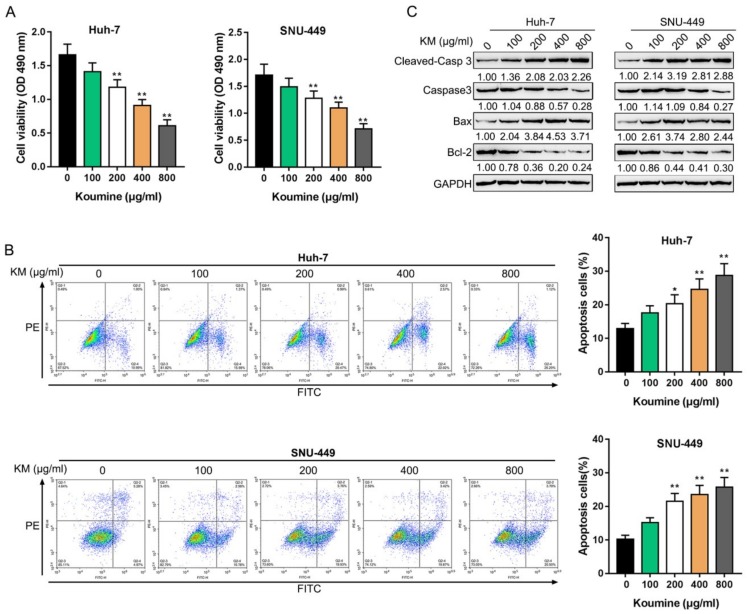
The killing effects of koumine on hepatocellular carcinoma (HCC) cells. HCC cells were treated with 0, 100, 200, 400, and 800 μg/mL koumine and examined for (**A**) cell viability by MTT assays; (**B**) cell apoptosis by flow cytometry; (**C**) protein levels of apoptosis-related factors, including cleaved-Caspase3, Caspase3, Bax, and Bcl-2. * *p* < 0.05, ** *p* < 0.01.

**Figure 2 biomolecules-09-00559-f002:**
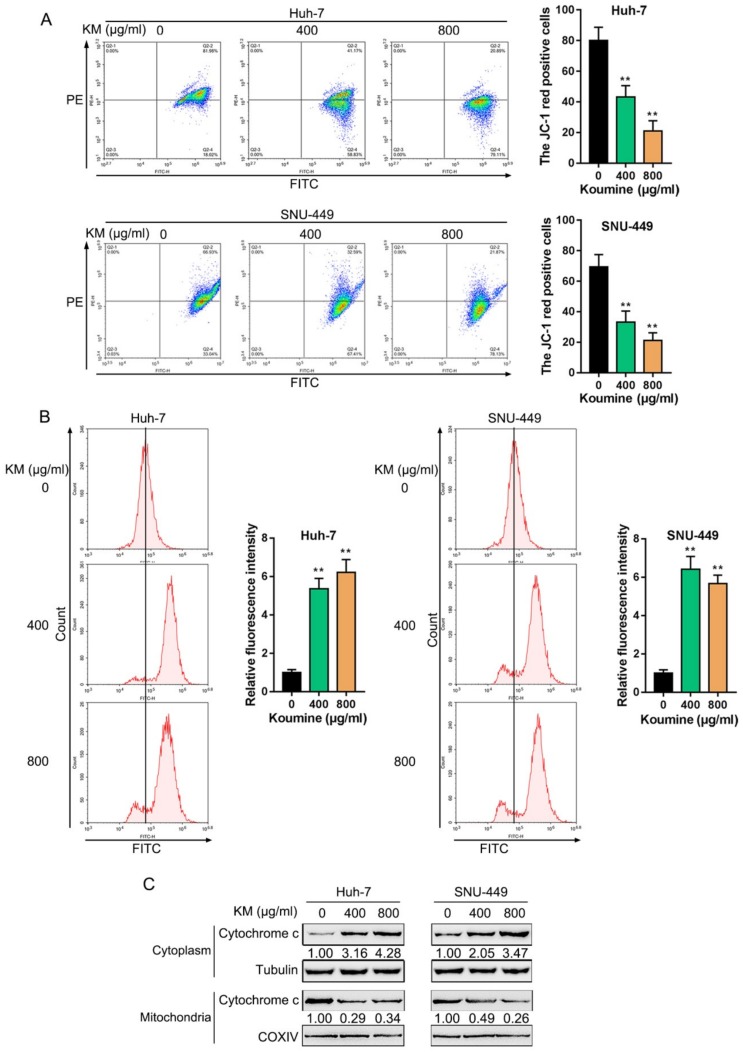
Effects of koumine on mitochondrial function in HCC cells. HCC cells were treated with 0, 400, and 800 μg/mL koumine and examined for (**A**) the mitochondrial membrane potential (ΔΨm) with the JC-1 mitochondrial transmembrane potential detection kit; (B) the reactive oxygen species (ROS) production by 2’,7’-Dichlorodihydrofluorescein diacetate (DCFH-DA) assay; (**C**) the protein level of cytochrome C by immunoblotting. * *p* < 0.05, ** *p* < 0.01.

**Figure 3 biomolecules-09-00559-f003:**
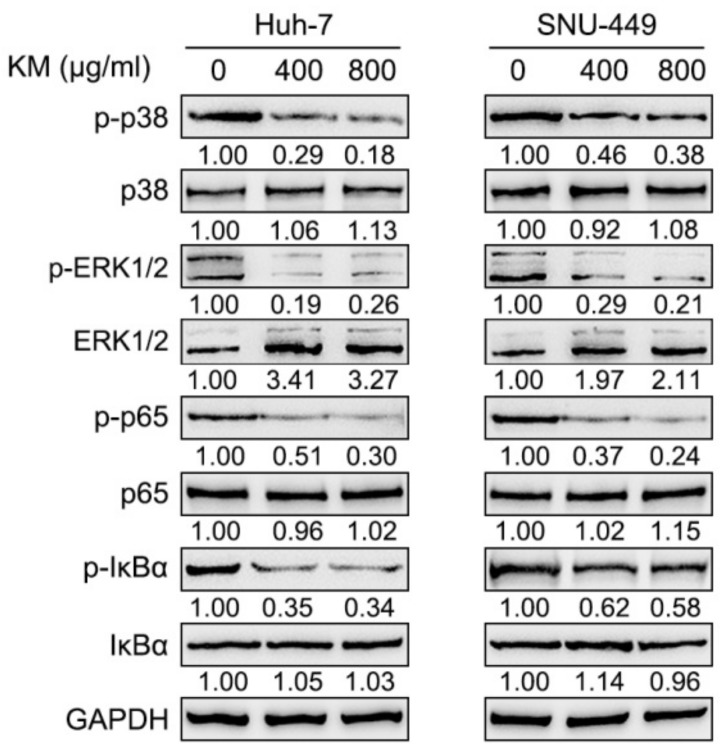
Koumine inhibits NF-κB and ERK/p38 MAPK signaling pathways in HCC cells. HCC cells were treated with 0, 400, and 800 μg/mL koumine and examined for the protein levels of p-ERK, ERK, p-p38, p38, p-p65, p65, p-IκBα, and IκBα.

**Figure 4 biomolecules-09-00559-f004:**
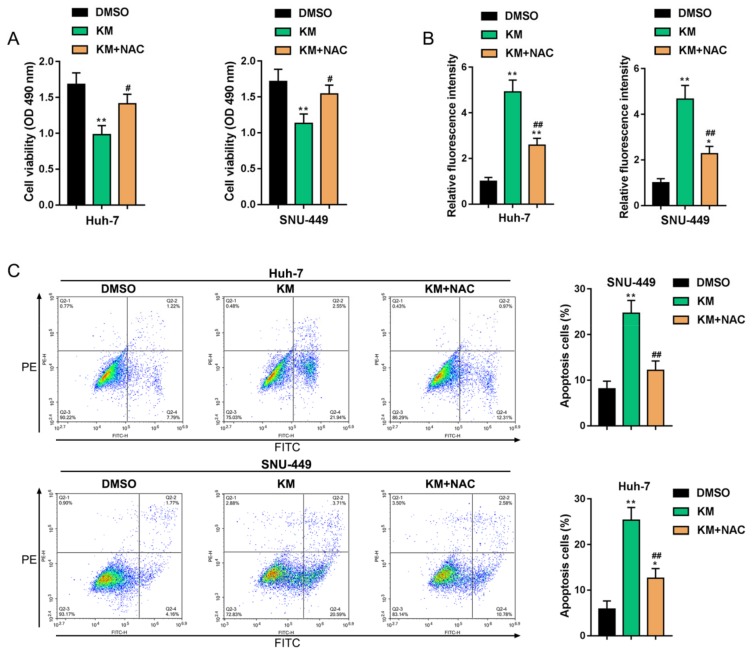
Koumine affects HCC cell behaviors through producing excess ROS. HCC cells were co-treated with 400 μg/mL koumine and 800 μM N-acetylcysteine (NAC) and examined for (**A**) cell viability by MTT assay; (**B**) the production of ROS by DCFH-DA assay; (**C**) cell apoptosis by flow cytometry. * *p* < 0.05 and ** *p* < 0.01 vs. DMSO control group, # *p* < 0.05 and ## *p* < 0.01 vs. KM group.

**Figure 5 biomolecules-09-00559-f005:**
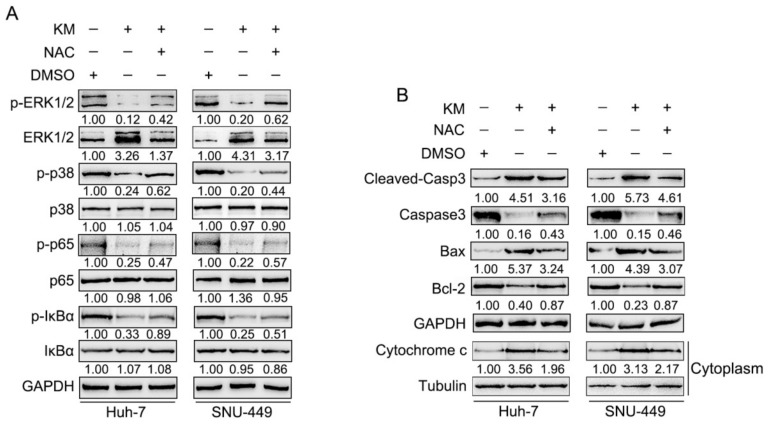
Koumine exerts its effects on HCC cells via ERK/p38 MAPK phosphorylation and NF-κB signaling. HCC cells were co-treated with 400 μg/mL koumine and 800 μM NAC and examined for (**A**) the protein levels of p-ERK, ERK, p-p38, p38, p-p65, p65, p-IκBα, and IκBα by immunoblotting; (**B**) the protein levels of apoptosis-related factors, including cleaved-Caspase3, Caspase3, Bax, and Bcl-2 by immunoblotting.

**Figure 6 biomolecules-09-00559-f006:**
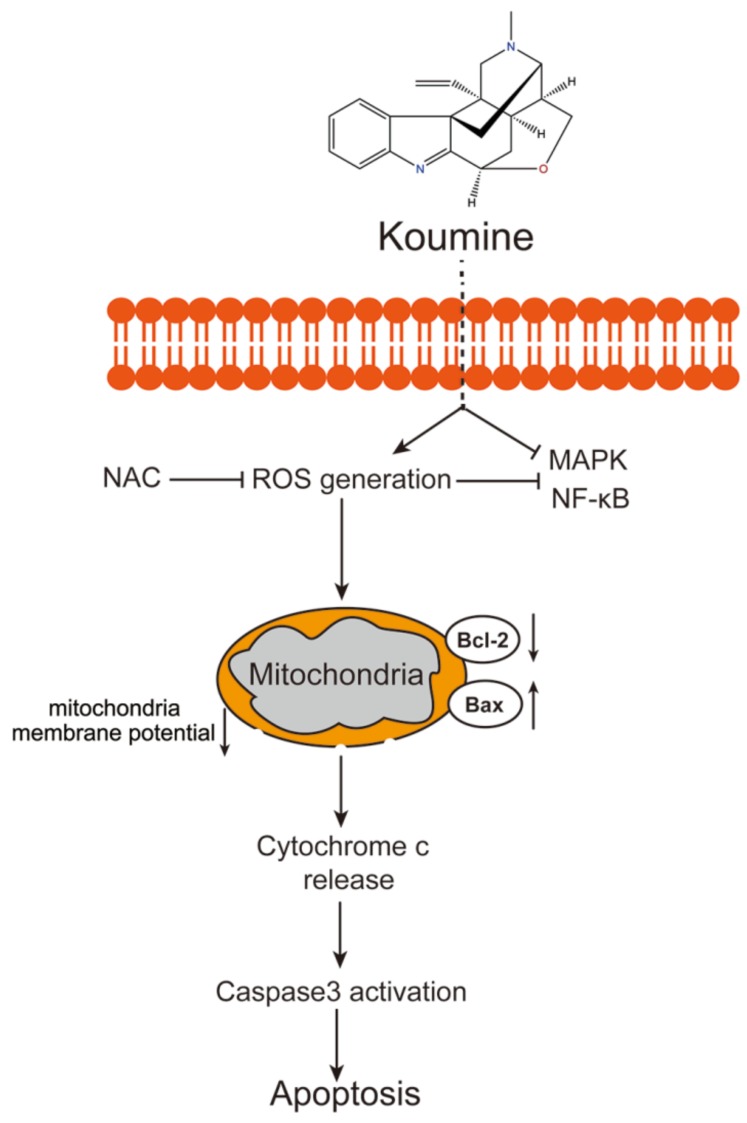
A schematic diagram of the proposed mechanisms of koumine induced apoptosis in HCC cells. The → indicates release, activation or induction.

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
