# Peer review of "Koumine Promotes ROS Production to Suppress Hepatocellular Carcinoma Cell Proliferation Via NF-κB and ERK/p38 MAPK Signaling"

_biomolecules, 2019, doi:10.3390/biom9100559_

Round 1

Reviewer 1 Report

The manuscript entitled “Koumine promotes the ROS production to suppress hepatocellular carcinoma cell proliferation via NF-κB 3 and ERK/p38 MAPK signaling” aims to demonstrate koumine contributes to regulating the signaling pathways of NF-κB and ERK/p38 MAPK within HCC through the excessive production of ROS, therefore inhibiting HCC cell proliferation and promoting HCC cell apoptosis.
Following recommends need modified:

1.The author uses the method of variance analysis to analyze the data so it is necessary to test the homogeneity of variance of the data.
2.The discussion is tedious. Author state much information which is irrelevant to this study so it need be modified.

Author Response

Comments and Suggestions for Authors

The manuscript entitled “Koumine promotes the ROS production to suppress hepatocellular carcinoma cell proliferation via NF-κB 3 and ERK/p38 MAPK signaling” aims to demonstrate koumine contributes to regulating the signaling pathways of NF-κB and ERK/p38 MAPK within HCC through the excessive production of ROS, therefore inhibiting HCC cell proliferation and promoting HCC cell apoptosis.

Following recommends need modified:

1.The author uses the method of variance analysis to analyze the data so it is necessary to test the homogeneity of variance of the data.

Reply:

Thank you for your professional advice. We tested the homogeneity of variance of all the data, and all comparison groups have the same finite variance.

2.The discussion is tedious. Author state much information which is irrelevant to this study so it need be modified.

Reply:

Thank you for your professional advice. We apologize for irrelevant information in the Discussion section. We deleted the irrelevant content and focus on the specific role and the underlying mechanism of koumine in HCC.

Reviewer 2 Report

In general, the manuscript is well written and the contents support their conclusion. I have some comments that I believe might help the authors in increasing the impact of this manuscript.

Comments
1. Is koumine not cytotoxic in normal cells? Please provide proof of this.
2. Do you think that ROS increased by the treatment of koumine originated from mitochondria? What is the author's view on this?
3. In Discussion part, please discuss the comparison with the results of previous studies (references, 21-23) in more depth.
4. What is the clinical significance of this study? Please give your views on this.

Author Response

Comments

Is koumine not cytotoxic in normal cells? Please provide proof of this.

Reply:

Thank you for your professional advice. We tested the effects of koumine on normal, healthy cells in our preliminary experiments (data not shown in the submitted manuscript). In the figure below, results from MTT assay showed that koumine could also suppressed the cell viability of liver cell line L02 and liver cancer cell line SNU-449. The cancer cell line SNU-449 showed more sensitive to koumine, compared to L02 (* p<0.05, ** p<0.01). The different effects of koumine on normal, healthy cells and cancer cells show that koumine might serve as a promising anti-tumor agent. However, these potential side-effects should be further evaluated and addressed in our future study.

Do you think that ROS increased by the treatment of koumine originated from mitochondria? What is the author's view on this?

Reply:

Thank you for your professional advice. We believe that koumine affects mitochondria-originated ROS. The conclusion that koumine affected mitochondrial production of ROS, therefore promoting the cancer cell apoptosis and inhibiting cancer cell viability was drawn based on koumine-induced changes in the mitochondrial membrane potential, the protein levels of cytochrome C, and the production of ROS.

In Discussion part, please discuss the comparison with the results of previous studies (references, 21-23) in more depth.

Reply:

Thank you for your professional advice. We deleted some irrelevant information and added more in-depth comparison with the previous studies. As follows:” As for the underlying mechanism, the anti-tumor effects of koumine have been attributed to its effects on mitochondria functions and mitochondrial production of ROS. Koumine not only lowers colorectal cancer LoVo cell membrane potential and mitochondrial membrane potential but also frees cytosolic calcium concentration, while increases ROS production and LoVo cell gap junction intercellular communication. Via the above-mentioned mechanisms, koumine induces LoVo cell apoptosis 22, 32. In breast cancer, koumine affected the apoptotic Caspase 3/Bcl-2 cascades to induce G2/M arrest and apoptosis in breast cancer MCF-7 cells 21. Herein, koumine can remarkably suppress the proliferation while promotes the apoptosis of HCC by enhancing cleaved-Caspase 3 and Bax protein levels whereas inhibiting Bcl-2 protein.”

What is the clinical significance of this study? Please give your views on this.

Reply:

Thank you for your professional advice. As we have mentioned, koumine slightly inhibited the viability of normal, healthy cells but significantly inhibited cancer cell viability. Thus, we believe that koumine may serve as a promising anti-tumor agent, which needs further in vivo and clinical investigations.
